# Characterization of Intestinal Microbiota in Lambs with Different Susceptibility to *Escherichia coli* F17

**DOI:** 10.3390/vetsci9120670

**Published:** 2022-12-01

**Authors:** Jingyi Sun, Weihao Chen, Zehu Yuan

**Affiliations:** 1College of Veterinary Medicine, Yangzhou University, Yangzhou 225009, China; 2College of Animal Science and Technology, Yangzhou University, Yangzhou 225009, China; 3Joint International Research Laboratory of Agriculture and Agri-Product Safety of Ministry of Education of China, Yangzhou University, Yangzhou 225009, China; 4International Joint Research Laboratory in Universities of Jiangsu Province of China for Domestic Animal Germplasm Resources and Genetic Improvement, Yangzhou University, Yangzhou 225009, China

**Keywords:** ETEC, *Escherichia coli* F17, challenge experiment, 16S rRNA, intestinal microbiota, lambs

## Abstract

**Simple Summary:**

*Escherichia coli* (*E. coli*) F17 is one of the major pathogenic bacteria responsible for diarrhea in farm animals; however, little is known about the effect of *E. coli* F17 infection on the intestinal microbiota. The aim of this study was to investigate the intestinal microbiota in lambs with different susceptibilities to *E. coli* F17. By conducting an *E. coli* F17 challenge experiment, lambs sensitive/resistant (SE/AN) to *E. coli* F17 were identified, and 16S rRNA gene sequencing was performed to evaluate the intestinal microbiota in SE/AN lambs. The results showed that a relatively higher level of richness and diversity were characterized in the bacterial communities in the AN lambs than in the SE lambs, while the abundance of *Lactococcus* and *Megasphaera elsdenii* was found to be significantly different between the AN and SE lambs. Furthermore, our results indicated that the *Bacteroidetes:Firmicutes* ratio can serve as a promising bacterial biomarker for *E. coli* F17 infection. Our results can help in the development of new insights for the treatment of farm animals infected by *E. coli* F17.

**Abstract:**

Diarrhea is one of the most commonly reported diseases in young farm animals. *Escherichia coli* (*E. coli*) F17 is one of the major pathogenic bacteria responsible for diarrhea. However, the pathogenicity of diarrhea in lambs involving *E. coli* F17 strains and how *E. coli* F17 infection modifies lambs’ intestinal microbiota are largely unknown. To evaluate diarrhea in newborn lambs with an infection of *E. coli* F17, 50 lambs were selected for challenge experiments and divided into four groups, namely, a high-dose challenge group, low-dose challenge group, positive control group, and negative control group. The *E. coli* F17 challenge experiments caused diarrhea and increased mortality in the experimental lamb population, with a higher prevalence (90%), mortality (35%), and rapid onset time (4–12 h) being observed in the high-dose challenge group than the results observed in the low-dose challenge group (75%, 10%, 6–24 h, respectively). After the challenge experiment, healthy lambs in the high-dose challenge group and severely diarrheic lamb in the low-dose challenge group were identified as lambs sensitive/resistant to *E. coli* F17 (*E. coli* F17 -resistant/-sensitive candidate, AN/SE) according to the histopathological detection. Results of intestinal contents bacteria plate counting revealed that the number of bacteria in the intestinal contents of SE lambs was 10^2~3^-fold greater than that of the AN lambs, especially in the jejunum. Then, 16S rRNA sequencing was conducted to profile the intestinal microbiota using the jejunal contents, and the results showed that SE lambs had higher *Lactococcus* and a lower *Bacteroidetes:Firmicutes* ratio and intestinal microbiota diversity in the jejunum than AN lambs. Notably, high abundance of *Megasphaera elsdenii* was revealed in AN lambs, which indicated that *Megasphaera elsdenii* may serve as a potential probiotic for *E. coli* F17 infection. Our study provides an alternative challenge model for the identification of *E. coli* F17-sensitive/-resistant lambs and contributes to the basic understandings of intestinal microbiota in lambs with different susceptibilities to *E. coli* F17.

## 1. Introduction

Among commonly reported diseases, diarrhea holds one of the most important places in young farm animals (lambs under two weeks old, calves younger than ten days, and newborn or just-weaned piglets) and is associated with complex microbial infections. Studies have shown that diarrheagenic *Escherichia coli* (*E. coli*) is the major pathogenic bacterium responsible for diarrhea [1]. Based on the virulence properties and clinical signs of host animals, diarrheagenic *E. coli* can be divided into six major pathotypes: enterotoxigenic *E. coli* (ETEC), enterohemorrhagic *E. coli* (EHEC), enteropathogenic *E. coli* (EPEC), enteroaggregative *E. coli* (EAEC), enteroinvasive *E. coli* (EIEC), and diffusely adherent *E. coli* (DAEC) [2]. Among them, ETEC is the major bacterial agent involved in young animal diarrhea. Mechanically, ETEC can adhere to the intestinal epithelium, leading to the production and secretion of enterotoxin via its fimbriae, thus causing severe disruption of the intestinal microbiota. The fimbrial adhesins, F5 [3], F17 [4], F18 [5], and F41 [6] are mainly associated with ETEC in calves, lambs, and piglets. Among them, *E. coli* F17 has been identified in diarrhea in animals in Asia [7,8], Europe [9,10], South America, [11] and North America [12] which indicates that *E. coli* F17 poses potential risks to farm animal globally. Considering the zoonotic characteristics and global distribution trend of *E. coli* F17, there is an urgent need to study the pathogenicity of diarrhea involving *E. coli* F17 strains and to reveal how *E. coli* F17 infection modifies intestinal microbiota.

Intestinal microbiota, composed of a large population of microorganisms, is often considered as a new “organ” in multiple biological progresses. Changes in the composition of intestinal microbiota are closely linked with diverse diseases, including inflammatory bowel disease [13], colorectal cancer [14], and obesity [15]. Recent studies in *Escherichia coli*-induced diarrhea have highlighted that changes in the intestinal microbiota composition exert differential effects on the intestinal susceptibility to ETEC infection, for example, high level of *Lactobacillus rhamnosus* can lead to increased serum IL-17A during *E. coli* F4 infection in piglets [16]; in humans, symptomatic ETEC infection is closely related to the outgrowth of *Enterobacteriaceae* [17]. However, studies on the change in microbiota of diarrheic animals caused by *E. coli* F17 infection remain scarce, especially in lambs.

Hence, our study was conducted to evaluate the effect of *E. coli* F17 infection on the intestinal microbiota in lambs. There were three components in the present study. Firstly, challenge experiments were established to examine the pathogenicity of the *E. coli* F17 strain in lambs. Secondly, histopathological examination experiments were performed to evaluate the pathogenicity to identify lambs with different susceptibility to *E. coli* F17 (*E. coli* F17-sensitive/-resistant). Finally, 16S rRNA sequencing was conducted to characterize the intestinal microbiota and taxonomic diversity of *E. coli* F17-sensitive/-resistant candidate lambs. Our results can provide a basic understanding of the intestinal microbiota in lambs with divergent susceptibility to *E. coli* F17 and may further provide support for the treatment of farm animals infected by the *E. coli* F17 strain.

## 2. Materials and Methods

### 2.1. Escherichia coli F17 Preparation

The *Escherichia coli* F17 strain (DN1401, fimbrial structural subunit: F17b, fimbrial adhesin subunit: subfamily II adhesins, originally isolated from diarrheic calves) was obtained from the School of Animal Medicine, Northeast Agricultural University. A single colony of *E. coli* F17 was grown on 50 mL of LB medium. A colony forming unit (CFU) was estimated by plate counting and stored at 4°C for the challenge experiment.

### 2.2. PCR Conditions for E. coli F17 Identification

The reaction mixture contained 10 µL 2× Taq PCR Master mix (TIANGEN BIOTECH Co., Ltd., Beijing, China), 0.8 µL each of forward and reverse primer, 2.0 µL DNA, and 6.4 µL RNase-free ddH_2_O (total volume, 20 μL). The PCR thermocycler program was performed following Bertin’s methods [18]. The PCR amplifications of the *E. coli* F17 were performed and amplified using Primer F (GGGCTGACAGAGGAGGTGGGGC) and Primer R (CCCGGCGACAACTTCATCACCGG).

### 2.3. Animals and Challenge Experimental Design

All of the experimental sheep were supplied by the Xilaiyuan Agriculture Co., Ltd. (Jiangsu Providence, China). One hundred healthy newborn Hu sheep lambs with similar weights (3 ± 0.5 kg) were randomly chosen and reared on lamb milk replacer free of antimicrobial additives and free of probiotics (Jingzhun^®^, Beijing Precision Animal Nutrition Research center, Beijing, China). All of the lambs were reared from one day old to three days old. Feces were collected from each lamb every 6 h for *E. coli* F17 identification to ensure that the lambs were free from *E. coli* F17 before the challenge experiment.

At three days after birth, 50 healthy lambs considered free from *E. coli* F17 (according to the PCR amplifications results of *E. coli* F17 using the collected feces) were randomly selected from the 100 aforementioned selected lambs and divided into four experimental groups, namely, the high-dose challenge group (20 lambs), low-dose challenge group (20 lambs), positive control group (five lambs), and negative control group (five lambs). The challenged dose was decided according to our previous research [19]. Each group was reared with strict separation and then the challenge experiments were conducted from four days old for up to seven days old. The detailed experimental design is shown in Table 1.

During the challenge experiment, fecal shedding of *E. coli* F17 was monitored by fecal sampling as described above, and the feces were recorded according to the Bristol stool form scale (Table 2). Only lambs with watery feces (Type 6, 7) were considered as diarrheic, and lambs with sausage-shaped feces were considered as healthy (Type 1, 2).

After the challenge experiments, severely diarrheic lambs in the low-dose challenge group, healthy lambs in the high-dose challenge group, and lambs in the positive/negative control group were slaughtered by euthanasia (KCl, 1 mg/kg) intravenous under deep anesthesia using pentobarbital sodium (1.5 mg/kg). About 100 mg of intestinal tissues (duodenum, jejunum, and ileum) and 3 mL of duodenum, jejunum, and ileum contents were collected and used for bacteria plate counting and histopathological examination following Wu’s method [20].

### 2.4. Jejunal Bacterial Community Sequencing

Six severely diarrheic lambs in the low-dose challenge group and six healthy lambs in the high-dose challenge group were finally chosen as *E. coli* F17-sensitive candidates (sensitive, SE) and *E. coli* F17-resistant candidates (antagonism, AN) according to the results of the bacteria plate counting and histopathological examination. The jejunum contents (2 mL) from six AN lambs and six SE lambs were collected for microbiota analyses and they were snap-frozen in liquid nitrogen and stored at −80 °C until use.

Total genome DNA was extracted from jejunum contents using the SDS method and diluted to 1 ng/µL using sterile water. The PCR amplifications of the 16S V3-V4 regions of the bacterial 16S rDNA gene were performed and amplified using universal Primer 341F (CCTAYGGGRBGCASCAG) and Primer 806R (GGACTACNNGGGTATCTAAT). Sequencing libraries were prepared using the TruSeq^®^ DNA PCR-Free Sample Preparation Kit (Illumina, San Diego, CA, USA) as per the manufacturer’s recommendations. All libraries were sequenced on the Illumina NovaSeq platform after purification and quantification. Sequencing work was conducted by a commercial sequencing provider (Beijing Novogene Technology Co., Ltd., Beijing, China).

### 2.5. 16s rRNA Data Analysis Process

Pair-end reads were merged using the FLASH tool (version 1.2.7) [21], and clean tags were generated after quality filtering per the QIIME quality-controlled process [22]. Then, the clean reads were aligned against the SILVA reference database using the UCHIME algorithm [23]. Chimeras were removed using the VSEARCH algorithm, and effective reads were finally generated [24]. The effective reads were assigned to the same operational taxonomic units (OTUs) with a 97% sequence similarity threshold by UPARSE software (version 7.0.1001) [25]. The taxonomy of OTUs were aligned against the SILVA [26] reference database (SILVA SSU 138) based on the Mothur algorithm [27].

After normalization of OTUs abundance information based on the minimum number of valid sequences, alpha diversity and beta diversity were estimated. For alpha diversity analysis, six indices were estimated including observed species, ACE, Chao1, Shannon, Simpson, and Good’s coverage using QIIME (version 1.9.1) [22].

For beta diversity analysis, the bacterial communities of different groups were statistically compared using the analysis of molecular variance (AMOVA) test [28]. A distance matrix among samples was measured using Jaccard and Bray–Curtis dissimilarity. Then, the results were visualized using principal coordinate analysis (PCoA) based on the WGCNA R library [29]. For the prediction of functional capabilities, the relative abundance of 16S rRNA data was analyzed using Tax4Fun [30] based on Kyoto Encyclopedia of Genes and Genomes (KEGG) orthologs.

### 2.6. Statistical Analyses

The data presented in our study were analyzed using SPSS 22.0 software. All data were presented as means ± standard error of the mean (SEM). Statistical analyses were performed following Ren’s method [31]. Significant differences were declared as the threshold of *p*-value < 0.05.

## 3. Results

### 3.1. Identification of E. coli F17

Before the challenge experiment, *E. coli* F17 was not detected in the lambs’ feces, which indicated that the experimental lambs were not infected by *E. coli* F17. During the challenge experiment, *E. coli* F17 was detected in the feces of lambs, which indicated that *E. coli* F17 was successfully established in the challenged lambs. (Figure 1)

### 3.2. Pathogenicity Record of E. coli F17 Challenge Experiment

Diarrheic feces were observed in *E. coli* F17-challenged lambs, with onset varying from 4 to 12 h post-challenge. After four days of the challenge experiment, 14/20 (75%), 18/20 (90%), 5/5 (100%), and 0/5 (0%) diarrheic lambs were observed in the high-dose challenge group, low-dose challenge group, positive control group, and negative control group, respectively. The prevalence of diarrhea and the death rate varied among the challenge group (Table 3). Details of the pathogenicity record of the *E. coli* F17 challenge experiment are presented in Appendix A.

### 3.3. Identification of Hu Sheep Sensitive and Resistant to the E. coli F17 Strain

Six severely diarrheic lambs in the low-dose challenge group and six healthy lambs in the high-dose challenge group were identified as candidate lambs sensitive and resistant to the *E. coli* F17 strain for the following experiment. Twelve candidate lambs sensitive/resistant to *E. coli* F17 and six lambs randomly chosen from the positive/negative control groups were slaughtered and their intestinal tissues (duodenum, jejunum, and ileum) and corresponding intestinal contents were collected to detect bacterial numbers. Details of selected lambs can be found in Appendix A.

### 3.4. Histopathological Examination Experiment

Diarrheic lambs exhibited severe diarrhea and pathological intestinal changes (Figure 2A). Moreover, severe pathological damage in the jejunum and ileum was observed. (Figure 3B,D,F). Healthy lambs displayed no obvious pathological changes (Figure 2A and Figure 3A,C,E).

### 3.5. Bacteria Plate Counting Result

The results of the bacteria plate counting experiment showed that the number of bacteria in the intestinal contents of the *E. coli* 17-sensitive candidates and the positive control group was numerically higher than that of the *E. coli* 17-resistant candidates and in the negative control group, the number of bacteria in the jejunum contents was numerically higher than that of the duodenum and jejunum contents (Table 4).

### 3.6. Jejunal Bacterial Communities Profile of Lambs with Different Susceptibility to Escherichia coli F17

Considering the susceptibility of the jejunum to *E. coli* F17 (based on the results of bacteria plate counting and the histopathological examination), the jejunum contents collected from *E. coli* F17-sensitive candidate lambs (sensitive group, SE) and *E. coli* F17-resistant candidate lambs (antagonism group, AN) were processed for 16s rRNA sequencing. After the quality filtering process, a total of 956,039 clean reads were generated from 12 samples (79,670 sequences per sample on average). Then, clean reads were clustered into 1115 OTUs and then assigned to 16 phyla, 31 classes, 79 orders, 127 families, 241 genera, and 163 species (Figure 4).

No significant differences (*p* > 0.05) were observed in the richness and diversity of the bacterial communities between the SE and AN group, but the jejunum contents of AN lambs had relatively higher richness and diversity values than those of SE lambs (Figure 5). Detailed richness and diversity information of each sample are presented in Appendix A.

An AMOVA test was performed to compare the beta diversity of different groups. No significant difference (*p* > 0.05) was observed in the bacterial communities between AN and SE. Jaccard and Bray–Curtis dissimilarity was calculated, and then, a PCoA plot was visualized. The results showed that all of the samples were clustered into two nearby branches (Figure 6).

Moreover, bacterial communities were measured between AN and SE groups. At the phylum level (Figure 7A), the AN samples were dominated by *Firmicutes* (41.43%), followed by *Proteobacteria* (21.52%), *Bacteroidota* (17.36%), and *Cyanobacteria* (14.20%). The SE samples were dominated by *Firmicutes* (43.47%), followed by *Proteobacteria* (16.66%), *Bacteroidota* (12.36%), and *Verrucomicrobiota* (11.08%). At the genus level (Figure 7B), the AN samples were dominated by unidentified *Chloroplast* (14.19%), followed by *Lactobacillus* (12.91%), *Bacteroides* (8.44%), and *Megasphaera* (7.08%). The SE samples were dominated by *Lactobacillus* (19.70%), followed by *Bacteroides* (8.84%), unidentified *Chloroplast* (6.77%), and *Escherichia-Shigella* (6.50%). At the species level (Figure 7C), the AN samples were mainly composed of *Megasphaera elsdenii* (7.08%), followed by *Comamonas kerstersii* (6.25%), *Veillonella magna* (5.54%), and *Bacteroides vulgatus* (3.60%). The SE samples were mainly composed of *Akkermansia muciniphila* (11.08%), followed by *Lactobacillus salivarius* (8.38%), *Lactobacillus agilis* (7.22%), and *Escherichia coli* (6.50%).

Notably, four bacterial communities closely related to intestinal health including: *Akkermansia muciniphila* bacterial species, *Lactobacillus salivarius* bacterial species, *Lactobacillus agilis* bacterial species, and *Escherichia coli* bacterial species in the SE samples were relatively more highly enriched than those of the AN samples (Figure 8). Moreover, the *Megasphaera elsdenii* bacterial species, *Comamonas kerstersii* bacterial species, and *Veillonella magna* bacterial species of the AN samples were relatively more highly enriched than those of the SE samples (Figure 9). Detailed results of differential enriched bacterial communities between the AN and SE samples can be found in Appendix A.

To further understand the biological function of intestinal microbiota in AN and SE samples, KEGG enrichment analyses were performed. In general, the bacterial communities in both AN and SE samples were enriched with similar functional categories. At KEGG level 1 (Figure 10A), the bacterial communities in all of the samples were mostly enriched in metabolism (45.23%), followed by genetic information processing (22.56%) and environmental information processing (14.37%). At KEGG level 2 (Figure 10B), the bacterial communities in all of the samples were mostly enriched in membrane transport (10.98%), followed by carbohydrate metabolism (10.57%) and replication and repair (9.24%). At KEGG level 3 (Figure 10C), the bacterial communities in all of the samples were mostly enriched in transport (6.93%), followed by DNA repair and recombination proteins (3.15%) and two component systems (2.73%). Detailed KEGG enrichment results can be found in Appendix A.

By comparing the enrichment results between AN and SE, some microbial biological functions were found to be significantly enriched in the AN samples, including lipid metabolism and metabolism of other amino acids at KEGG level 2 (Figure 11A) and propanoate metabolism, fatty acid biosynthesis, and inositol phosphate metabolism at KEGG level 3 (Figure 11B).

Conversely, endocrine and metabolic diseases at KEGG level 2 (Figure 11A) and mitochondrial biogenesis, nucleotide excision repair, selenocompound metabolism, thiamine metabolism, vancomycin resistance, tuberculosis, ferroptosis, and primary bile acid biosynthesis at KEGG level 3 (Figure 11B) were found to be significantly enriched in the SE samples.

## 4. Discussion

### 4.1. Effect of E. coli F17 on Lambs

*E. coli* F17, an ETEC phenotype of the *Escherichia coli* family, is associated with high morbidity and mortality [32] in young farm animals. The global prevalence of *E. coli* F17 causing diarrhea provides a renewed sense of urgency for *E. coli* F17 research. However, remarkably little research has been reported to evaluate the pathogenicity and clinical signs of *E. coli* F17.

Previous reports have well demonstrated challenge experiments in mice via multiple kinds of ETEC; although the experimental mice developed diarrhea, no pathogenic *E. coli* were detected in the feces of the mice, which indicated that this strategy is not feasible and ST-producing strains cannot be identified in mice [33]. Hence, our study considered the following points: firstly, newborn lambs were used as model animals to simulate diarrhea caused by the *E. coli* F17 strain; secondly, the lambs were fed with lamb milk powder free of antibiotics and probiotics to avoid passive immunity [34]; thirdly, identification of *E. coli* F17 in lamb feces was conducted before and after the first challenge to exclude the possibility of the experimental lambs being naturally infected with *E. coli* F17. The results of *E. coli* F17 identification showed that experimental lambs were only infected by the challenged *E. coli* F17 strain, which indicates the effectiveness and reliability of our challenge experiment.

The pathogenicity of *E. coli* F17 strain was assessed in newborn lambs via two challenge groups (high/low dose). The results showed that regardless of the challenge dose, the health of the challenged lambs was severely affected and the mortality increased. Compared to the low-dose challenge group, relatively higher prevalence and mortality were observed in the high-dose challenge group. Additionally, most of the lambs were observed with diarrhea within 4–12 h after the first challenge, which was much faster than lambs in the low-dose challenge group. Therefore, it is evident that a higher challenge dose of *E. coli* F17 led to obvious changes in the onset time, prevalence, and mortality. Notably, dying lambs, especially in the high-dose challenge group, were perceived to have difficulty breathing, cervical spine stiffness, and usually died within 4–8 h. Previous research reported similar complications in ETEC challenge experiments including septicemia and edema [32]. Collectively, we conclude that *E. coli* F17 may potentially be a causative agent for multiple complex complications in this study.

Histomorphological features of the intestine are important indicators of gut health in animals [35]. Multiples reports [36,37,38] have demonstrated the damage of ETEC infections to the intestinal villi and crypt. A previous study on *E. coli* F18 suggested that *E. coli*-resistant and -sensitive individuals can be identified via challenge experiments in piglets [20]. In the present study, six severely diarrheic lambs in the low-dose challenge group and six healthy lambs in the high-dose challenge group were identified as candidate lambs for *E. coli* F17-resistant and -sensitive individuals. Results of the histopathological examination and pathological tissue section showed that severe diarrhea and pathological intestinal damage were observed in SE lambs.

### 4.2. Intestinal Microbiota in Lambs with Different Susceptibility to E. coli F17

The intestinal microbiota is imperative for immune system development, and the health of newborn animals is largely characterized by balanced intestinal microbiota [39]. The plate counting experiment was conducted to evaluate the number of bacteria in the intestinal contents of candidate lambs, and it is evident that different challenge models lead to changes in the intestinal bacteria. Consistent with the previous research on *E. coli* F18 [20], the number of bacteria in the intestinal contents of diarrheic individuals (*E. coli* F17-sensitive candidates and the positive control group) was 10^2~3^-fold greater than that of healthy individuals (*E. coli* F17-resistant candidates and the negative control group), especially in the jejunum.

Considering the susceptibility of the jejunum to *E. coli* F17 (based on the results of bacteria plate counting and the histopathological examination), the jejunum contents collected from *E. coli* F17-sensitive candidate lambs (sensitive group, SE) and *E. coli* F17-resistant candidate lambs (antagonism group, AN) were processed for 16s rRNA sequencing.

Our results revealed a relatively higher level of richness and diversity in the bacterial communities in the AN lambs than in the SE lambs. Similar findings have also been reported by Rhouma et al. [40] and Peng et al. [41], who demonstrated a shape decrease in piglets fecal and jejunal microbiota alpha diversity after an ETEC challenge. These results suggest that the relatively stable intestinal microbiota may play a resistant role to diarrhea after being exposed to diarrheagenic *E. coli*. Thus, it seems that a microbial balance might protect AN lambs against diarrhea caused by *E. coli* F17. However, inconsistent with the results in previous reports [42,43,44], no significant differences in bacterial communities were observed between AN and SE lambs. PCoA analysis results also showed that AN and SE lambs were clustered into two nearby branches. This divergence was likely due to all of the experimental lambs (AN and SE lambs) for 16s rRNA sequencing being challenged with *E. coli* F17 in our study, while the significant differences in microbiota were initially revealed between the challenged and unchallenged individuals.

Previous studies have shown that ETEC-induced diarrhea is associated with a decrease in the *Bacteroidetes:Firmicutes* ratio [45,46,47]. In the present study, a lower ratio of *Bacteroidetes:Firmicutes* was also revealed in SE lambs (0.28) than in AN lambs (0.42) in the phylum level. Therefore, our findings support the idea that the *Bacteroidetes:Firmicutes* ratio can serve as a promising biomarker for ETEC infection. At the species level, the abundance of *Akkermansia muciniphila*, *Lactobacillus salivarius*, *Lactobacillus agilis*, and *Escherichia coli* in SE lambs was relatively higher than that of AN lambs. Akkermansia, which belongs to *Verrucomicrobia*, has been proven to have a positive effect on intestinal health [48]. Li et al. reported that the high abundance of *Akkermansia* can alleviate ETEC K88-induced oxidative damage in mice [42]. Peng et al. [41] reported a potential association between the recovery from ETEC-induced diarrhea and the abundance of *Lactobacillus* in piglets’ jejunum including *Lactobacillus amylovorus, Lactobacillus acidophilus,* and *Lactobacillus crispatus.* Taken together, our results suggested that *Akkermansia muciniphila*, *Lactobacillus salivarius*, and *Lactobacillus agilis* might contribute to the recovery from *E. coli* F17-induced diarrhea in SE lambs. Of course, in-depth studies are needed to confirm our ideas.

In AN lambs, high abundance of *Megasphaera elsdenii*, *Comamonas kerstersii*, and *Veillonella magna* was revealed. It is noteworthy that the revealed AN:SE ratio in the abundance of *Megasphaera elsdenii* was over 7000. As previously reported, the challenge with *E. coli* F18 [49] and K88 [50] can reduce the relative abundance of *Megasphaera elsdenii* in pigs. Therefore, similar alterations in the abundance of *Megasphaera elsdenii* of lambs in this study may contribute to an important role rather than a simple increase in the abundance of *Megasphaera elsdenii* in the susceptibility of lambs to *E. coli* F17. *Comamonas kerstersii* is a Gram-negative bacterium closely related to human abdominal and urinary tract infections, and bacteraemia and could be an opportunistic pathogen in humans [51]. *Veillonellae* are the most prevalent and predominant bacteria in the oral and gastrointestinal tract microbiota. In rare cases, *Veillonella* can cause serious infections such as meningitis, endocarditis, and osteomyelitis [52]. The specific roles of *Comamonas kerstersii* and *Veillonella magna* in ETEC-induced diarrhea are still largely unknown; however, a higher abundance found in the present AN lambs than in the SE lambs may represent a potential predisposing factor for susceptibility to *E. coli* F17.

For an in-depth understanding of the biological function of intestinal microbiota in response to *E. coli* F17 infection, biological function enrichment analyses were performed. The results revealed that metabolic-related KEGG pathways such as metabolism, genetic information processing, and membrane transport were enriched with the highest abundance. No significant change was revealed in the top enriched pathways between AN and SE lambs. This finding suggested that intestinal microbiota primarily play important roles in nutrient metabolism, both in AN and SE lambs.

Compared to the pathways enriched in AN lambs, nine pathways including endocrine and metabolic diseases, nucleotide excision repair, vancomycin resistance, tuberculosis, ferroptosis, and primary bile acid biosynthesis were more significantly enriched in SE lambs. These results provided the implication that in SE lambs, the intestinal microbiota was closely associated with diseases and these results could be attributed to the infection of *E. coli* F17. On the other hand, five metabolism-related KEGG pathway were also significantly enriched in AN lambs, which indicates that a stable intestinal metabolism might be responsible for the susceptibility to *E. coli* F17 in AN lambs.

## 5. Conclusions

A challenge experiment with *E. coli* F17 strains was successfully established. Regardless of the challenge dose, *E. coli* F17 increased the mortality and characteristic lesions in the intestine. The results of the histopathological examination demonstrate that resistance differences to *E. coli* F17 exist in lambs and they are closely associated with the alteration of intestinal microbiota. Lower microbiota diversity in the jejunum was revealed in SE lambs. Our results also indicated the abundance of bacterial communities (e.g., *Bacteroidetes:Firmicutes* ratio and *Megasphaera elsdenii*) that are closely associated with the susceptibility of lambs to *E. coli* F17. As such, these findings can contribute to our understanding of *E. coli* F17 infection and likewise provide an important database for the identification of *E. coli* F17-resistant/-sensitive individuals for further research.

## Figures and Tables

**Figure 1 vetsci-09-00670-f001:**
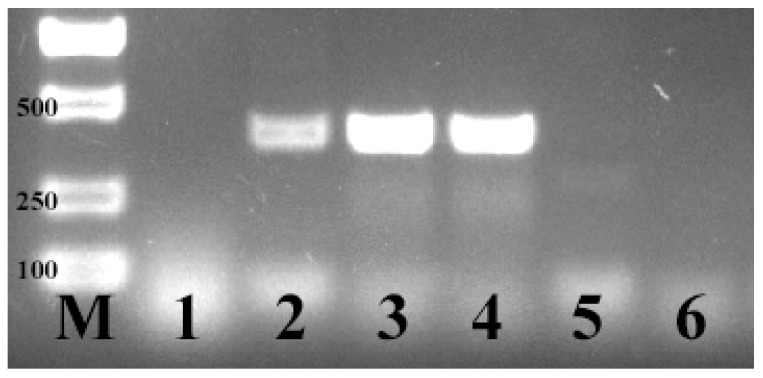
PCR detection of *E. coli* F17. M: DL 2000 marker; 1: ddH_2_O; 2: *E. coli* F17; 3,4: lambs’ feces collected during challenge experiment; 5,6: lambs’ feces collected before challenge experiment.

**Figure 2 vetsci-09-00670-f002:**
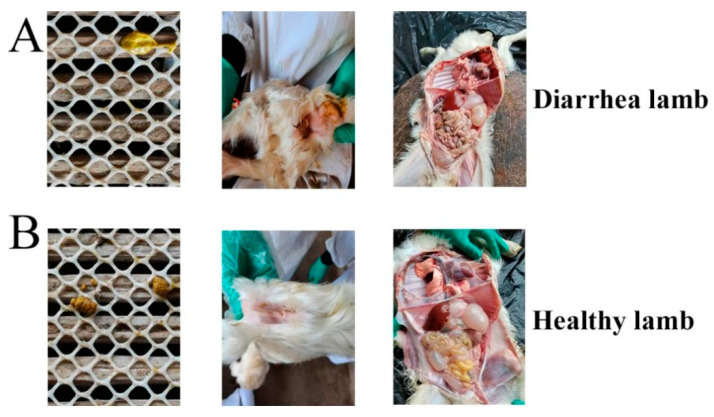
Histopathological detection of diarrheic lambs (**A**) and healthy lambs (**B**).

**Figure 3 vetsci-09-00670-f003:**
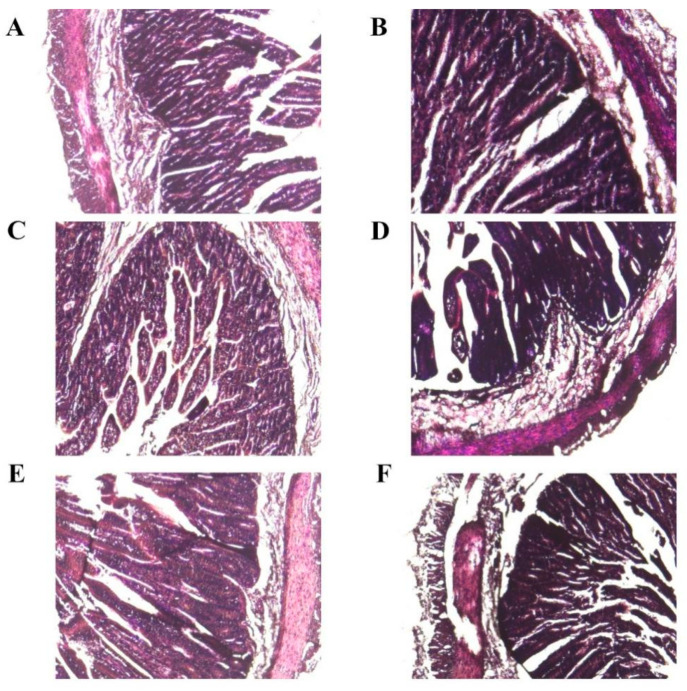
Pathological tissue section of healthy (**A**,**C**,**E**) lambs and diarrheic (**B**,**D**,**F**) lambs. (40×). (**A**,**B**) Duodenum; (**C**,**D**) jejunum; (**E**,**F**) ileum.

**Figure 4 vetsci-09-00670-f004:**
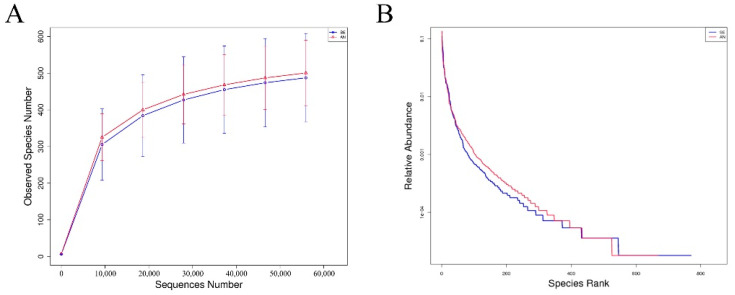
Observed species (**A**) and rank abundance (**B**) of identified OTUs in AN (red) and SE (blue) samples.

**Figure 5 vetsci-09-00670-f005:**
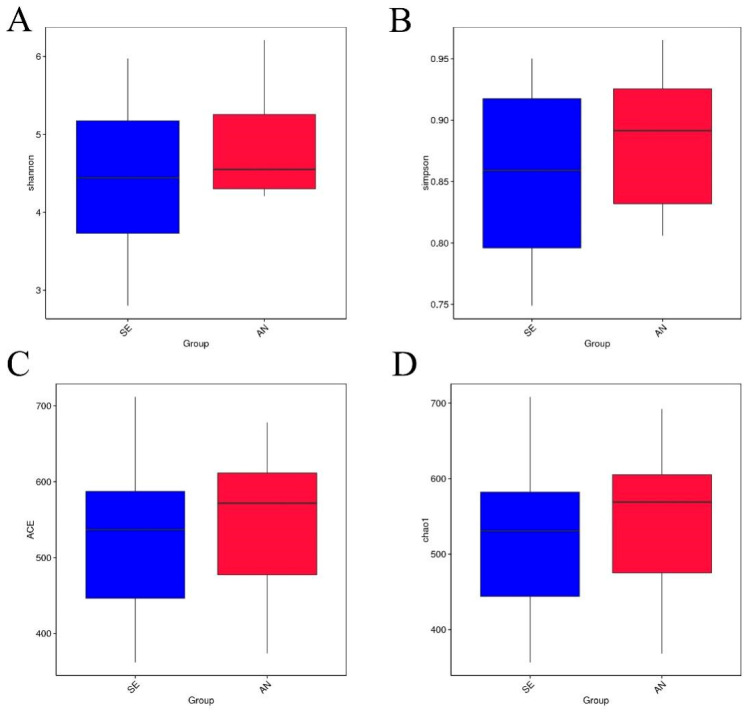
Comparison of alpha diversity indices of the bacterial communities in AN (red) and SE (blue) samples. (**A**) Comparison of the Shannon index. (**B**) Comparison of the Simpson index. (**C**) Comparison of the Chao1 index. (**D**) Comparison of the ACE index.

**Figure 6 vetsci-09-00670-f006:**
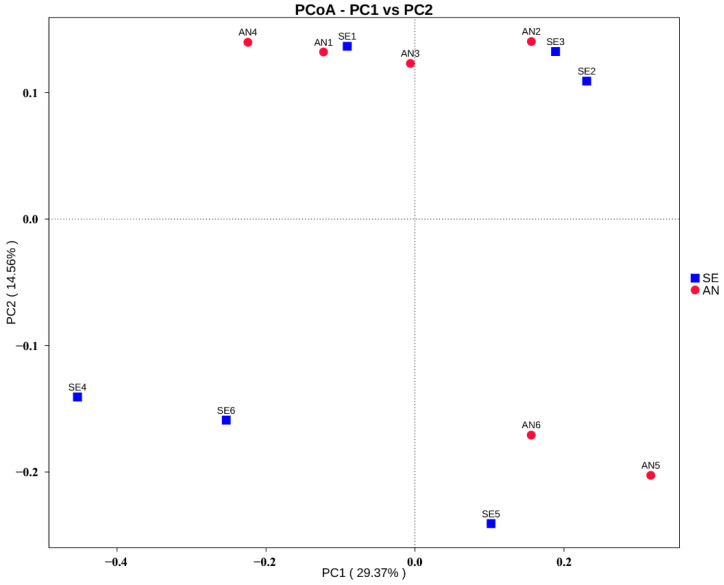
Principal coordinate analysis (PCoA) plot of the microbiota structure between AN (red) and SE (blue) samples.

**Figure 7 vetsci-09-00670-f007:**
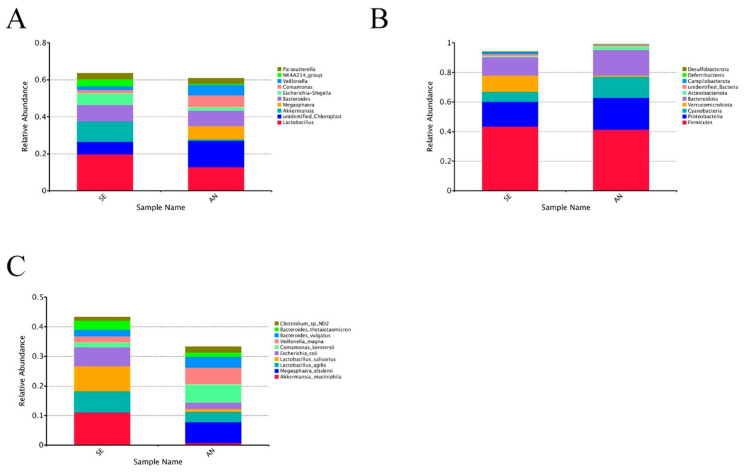
Distribution of the top bacterial communities in AN and SE samples at the phylum level (**A**), genus level (**B**), and the species level (**C**).

**Figure 8 vetsci-09-00670-f008:**
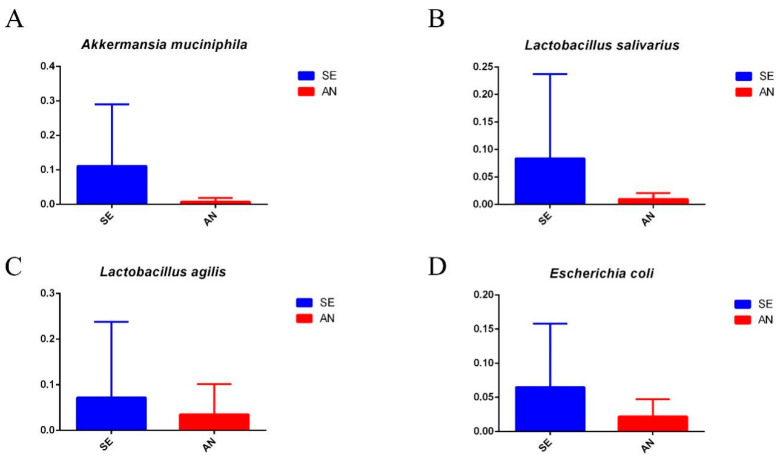
Relative abundance of the *Akkermansia muciniphila* bacterial species (**A**), *Lactobacillus salivarius* bacterial species (**B**), *Lactobacillus agilis* bacterial species (**C**), and *Escherichia coli* bacterial species (**D**) between the AN and SE samples.

**Figure 9 vetsci-09-00670-f009:**
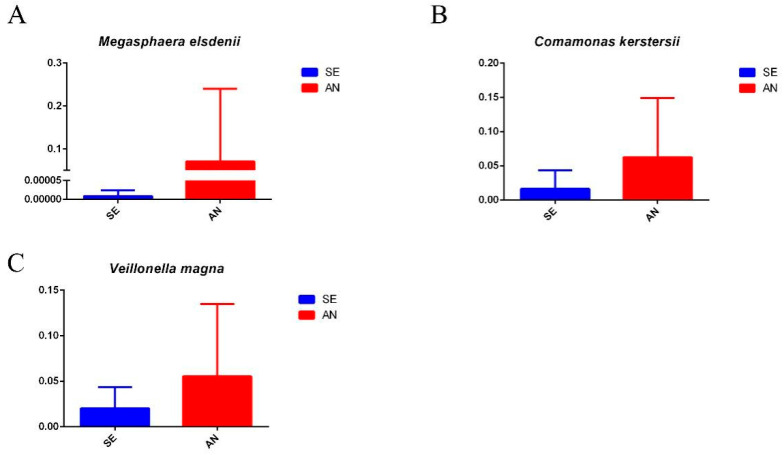
Relative abundance of the *Megasphaera elsdenii* bacterial species (**A**), *Comamonas kerstersii* bacterial species (**B**), and *Veillonella magna* bacterial species (**C**) between the AN and SE samples.

**Figure 10 vetsci-09-00670-f010:**
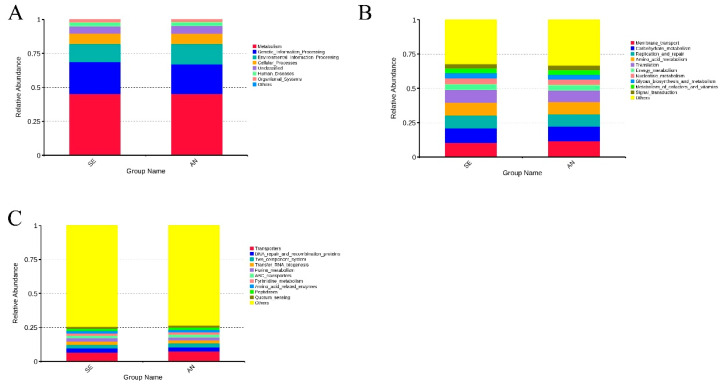
Top 10 enriched functional categories (KEGG level 1 (**A**), level 2 (**B**), and level 3 (**C**)) of the bacterial communities in the AN and SE samples.

**Figure 11 vetsci-09-00670-f011:**
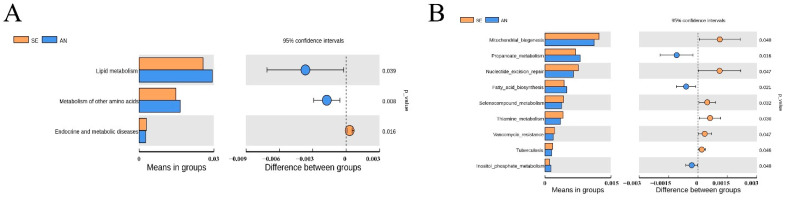
Differently enriched microbial biological functions between AN and SE samples at KEGG level 2 (**A**) and level 3 (**B**).

**Table 1 vetsci-09-00670-t001:** *E. coli* F17 strains for different challenge groups.

Group	No. of Lambs	Treatment	Dose (CFU)
High-dose challenge group	20	*E. coli* F17	5 × 10^9^
Low-dose challenge group	20	*E. coli* F17	5 × 10^8^
Positive control group	5	*E. coli* F17	1 × 10^10^
Negative control group	5	LB medium	-

Abbreviations: CFU, colony forming unit.

**Table 2 vetsci-09-00670-t002:** Bristol Stool Form Scale.

Type	Form
Type 1	Separate hard lumps, like nuts
Type 2	Sausage-shaped but lumpy
Type 3	Like a sausage or snake but with cracks on its surface
Type 4	Soft blobs with clear-cut edges
Type 5	Like a sausage or snake, smooth and soft.
Type 6	Fluffy pieces with ragged edges, a mushy stool
Type 7	Watery, no solid pieces

**Table 3 vetsci-09-00670-t003:** Pathogenicity Record of *E. coli* F17 challenge experiment.

Group	No. of Challenged Lambs	Prevalence	Death Rate
Low-dose challenge group	20	14/20, 75%	2/20, 10%
High-dose challenge group	20	18/20, 90%	7/20, 35%
Positive control group	5	5/5, 100%	3/5, 60%
Negative control group	5	0/5, 0%	0/5, 0%

Note: the number of dead lambs was included in diarrheic lambs.

**Table 4 vetsci-09-00670-t004:** Comparison of bacterial numbers in lambs in different groups.

10 µL Coated Plate	Intestinal Tract	Dilution Multiple				Average Count of Bacterial (CFU/mL)
		10^4^	10^5^	10^6^	10^7^	
sensitive candidates	duodenum	>1000	>500	128	13	1.29 × 10^8^
	jejunum	>1000	>500	176	19	1.83 × 10^8^
	ileum	>1000	>500	120	7	0.95 × 10^8^
resistant candidates	duodenum	113	5	NG	NG	8.20 × 10^5^
	jejunum	>500	170	9	NG	1.30 × 10^6^
	ileum	119	8	NG	NG	9.95 × 10^5^
positive control group	duodenum	>1000	>500	158	8	1.19 × 10^8^
	jejunum	>1000	>500	241	11	1.76 × 10^8^
	ileum	>1000	>500	147	3	0.89 × 10^8^
negative control group	duodenum	70	6	NG	NG	6.50 × 10^5^
	jejunum	64	NG	NG	NG	6.40 × 10^5^
	ileum	336	27	NG	NG	3.03 × 10^5^

Note: NG represents no growth.

## Data Availability

The raw sequencing datasets presented in this study can be found in online repositories: https://www.ncbi.nlm.nih.gov/ (accessed on 15 April 2022), PRJNA827002.

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
