# Peer review of "Characterization of Intestinal Microbiota in Lambs with Different Susceptibility to Escherichia coli F17"

_vetsci, 2022, doi:10.3390/vetsci9120670_

Round 1

Reviewer 1 Report

INTRODUCTION: The literature review should be expanded and updated. It would be necessary to focus more specifically on E.coli in lambs if possible. The classification of the six different pathotypes of E.coli, although rightly cited, should be updated, as recent studies in other species show that in many cases pathogenesis depends on a multiplicity of factors and not only on the pathotype of E.coli.

No mention is made in this section of previous studies focusing on gut microbiota, even in other species. Considering that there is a good deal of results focused on the different bacterial populations found, it should introduce the basic notions about gut microbiota and the different genera that can be found and whether these can be considered beneficial or less so.

MATERIALS AND METHODS: 

-For the PCR conditions, it is not necessary to have a specific table for the primers, as they can be entered in the text.

- no mention is made of the method of final selection of 50 animals considered free of E.coli F17, how are these animals selected from the initial group of 100 animals? is this a random choice or does it respond to any specific guidelines?

RESULTS:

- About the identification of sensitive and resistant animals, I understand that 6 animals form the low-dose challenge group and other 6 animals from the high-dose challenge group have been chosen, but I cannot find informations about how the 6 lambs in positive/negative control groups have been chosen (how many animals from the positive group and how many from the negative one?)

Author Response

Reviewer 1

INTRODUCTION: The literature review should be expanded and updated. It would be necessary to focus more specifically on E.coli in lambs if possible. The classification of the six different pathotypes of E.coli, although rightly cited, should be updated, as recent studies in other species show that in many cases pathogenesis depends on a multiplicity of factors and not only on the pathotype of E.coli.

No mention is made in this section of previous studies focusing on gut microbiota, even in other species. Considering that there is a good deal of results focused on the different bacterial populations found, it should introduce the basic notions about gut microbiota and the different genera that can be found and whether these can be considered beneficial or less so.

 Response: Thanks for the reviewer’s comments. A recent study has been cited on different pathotypes of E. coli, besides, the introduction of recent advance focusing on the intestinal microbiota has been added, details can be seen in our revised manuscript, line 61-72.

MATERIALS AND METHODS: 

-For the PCR conditions, it is not necessary to have a specific table for the primers, as they can be entered in the text.

Response: Thanks for the reviewer’s comments. We have revised as request, details can be found in revised manuscript, line 84-89.

- no mention is made of the method of final selection of 50 animals considered free of E.coli F17, how are these animals selected from the initial group of 100 animals? is this a random choice or does it respond to any specific guidelines?

Response: Sorry for the confusion we made, approximately 85 healthy lambs considered free of E.coli F17 (according to the PCR amplifications results of the E. coli F17 using collected feces) were identified on 3 days after birth, 50 lambs were randomly chosen form the 85 healthy lambs.

The sentence has been revised as:’ At three days after birth, 50 healthy lambs considered free from E. coli F17 (according to the PCR amplifications results of the E. coli F17 using collected feces) were randomly selected from the aforementioned selected lambs and divided into four experimental groups……’, details can be seen in revised manuscript, line 98-101.

RESULTS:

- About the identification of sensitive and resistant animals, I understand that 6 animals form the low-dose challenge group and other 6 animals from the high-dose challenge group have been chosen, but I cannot find information about how the 6 lambs in positive/negative control groups have been chosen (how many animals from the positive group and how many from the negative one?)

Response: Sorry for the confusion we made, 3 lambs from the positive control group and 3 lambs from the negative control group were randomly chosen, the sentence has been revised as:’ Twelve candidate lambs sensitive/resistant to E. coli F17 and 6 lambs randomly chosen from the positive/negative control groups were slaughtered’. The chosen lambs were also highlighted in supplementary table 1.

Reviewer 2 Report

This manuscript provides an insight into the intestinal microbiota of lambs with different susceptibility to E.coli F17. The experiment is well designed and several candidate bacteria related to E.coli F17 were identified,which may further aim the treatment of E.coli F17 infection in farm livestocks. However, several minor question should be addressed before the manuscript can be accepted, the specific comments are list as below

Line 88: Why do the author performe the challenge experiment at 4-day-old instead of starting challenge immediately after birth. 

Line 100:delete '-' 

Line 133: Please provide the detailed version of used software. 

Line 176: Change 'lambs challenged' to the number of challenged lambs. 

Line 218: The authors identified that the number of bacteria were higher in SE lambs than that in AN lambs. Why the diversity of bacterial community showed similar pattern between AN and SE lambs 

Line 299:Did the authors examined the gene expression profile of AN and SE lambs, is there any identified correlation between the DEGs and the differental bacterias?

Author Response

Reviewer 2

This manuscript provides an insight into the intestinal microbiota of lambs with different susceptibility to E.coli F17. The experiment is well designed and several candidate bacteria related to E.coli F17 were identified,which may further aim the treatment of E.coli F17 infection in farm livestocks. However, several minor question should be addressed before the manuscript can be accepted, the specific comments are list as below:

Response: We thank you very much for your positive comments on our manuscript.

Line 88: Why do the author perform the challenge experiment at 4-day-old instead of starting challenge immediately after birth.

Response: All lambs were reared on lamb milk replacer for the first three days after birth. to ensure the experimental lambs were not infected by E.coil F17 in the natural environment and decrease the death rate of experimental lambs (cause most newborn lambs will die after the challenge experiment, if they start immediately after birth.).

Line 100: delete '-'

Response: Thanks for the reviewer’s correction. We have revised the sentence as request.

Line 133: Please provide the detailed version of used software.

Response: Thanks for the reviewer’s comments, we have added the version of used software, details can be seen in the revised manuscript, line 133-151.

Line 176: Change 'lambs challenged' to the number of challenged lambs.

Response: Thanks for the reviewer’s correction. We have revised the sentence as request.

Line 218: The authors identified that the number of bacteria were higher in SE lambs than that in AN lambs. Why the diversity of bacterial community showed similar pattern between AN and SE lambs?

Response: This divergence was likely due to all experimental lambs for 16s rRNA sequencing were all challenged with E. coli F17 in our study, hence, the diversity of bacterial community in AN and SE lambs were similar. Although no significant difference was identified between AN and SE lambs, the jejunum contents of AN lambs had relatively higher richness and diversity values than those of SE lambs. Besides, the domain bacterial communities of AN and SE lambs were clearly different.

Line 299: Did the authors examined the gene expression profile of AN and SE lambs, is there any identified correlation between the DEGs and the differential bacteria?

Response: In addition to 16s rRNA seq, RNA-seq and LC-MS were also performed to study the transcriptomic and metabolic profiles (unpublished data) of AN and SE lambs, subsets of potential candidate genes and metabolites were identified (i.e. TFF2, LYPD8, FAHFAs, propionylcarnitine). Besides, an integrated omics analysis was also performed to gain insight into the crosstalk between the identified bacteria, genes, metabolites. (unpublished data)

Details can be seen in our previously published paper:

  1. Chen W, Lv X, Zhang W, et al. (2022) Insights Into Long Non-Coding RNA and mRNA Expression in the Jejunum of Lambs Challenged With Escherichia coli F17. Front. Vet. Sci. 9:819917. doi: 10.3389/fvets.2022.819917
  2. Chen,W.; Lv, X.; Zhang,W.; et al. Non-Coding Transcriptome Provides Novel Insights into the Escherichia coli F17 Susceptibility of Sheep Lamb. Biology 2022, 11, 348. https:// doi.org/10.3390/biology11030348
